# Soluble CD163-Associated Dietary Patterns and the Risk of Metabolic Syndrome

**DOI:** 10.3390/nu11040940

**Published:** 2019-04-25

**Authors:** Tzu-Yu Hu, Shin-Yng Lee, Chun-Kuang Shih, Meng-Jung Chou, Meng-Chieh Wu, I-Chun Teng, Chyi-Huey Bai, Nindy Sabrina, Alexey A. Tinkov, Anatoly V. Skalny, Jung-Su Chang

**Affiliations:** 1School of Nutrition and Health Sciences, College of Nutrition, Taipei Medical University, Taipei 11031, Taiwan; ma07106003@tmu.edu.tw (T.-Y.H.); shinyng90@gmail.com (S.-Y.L.); ckshih@tmu.edu.tw (C.-K.S.); ma07106005@tmu.edu.tw (M.-J.C.); ma07107007@tmu.edu.tw (M.-C.W.); eva850602@gmail.com (I.-C.T.); 2Department of Public Health, College of Medicine, Taipei Medical University, Taipei 11031, Taiwan; baich@tmu.edu.tw; 3Department of Public Health, College of Public Health, Taipei Medical University, Taipei 11031, Taiwan; 4Nutrition Program, Faculty of Food Technology and Health, Sahid Jakarta University, Jakarta 12870, Indonesia; sabrina.nindy@gmail.com; 5Peoples’ Friendship University of Russia (RUDN University), Miklukho-Maklaya St., 6, Moscow 105064, Russia; tinkov.a.a@gmail.com (A.A.T.); skalnylab@gmail.com (A.V.S.); 6Laboratory of Biotechnology and Applied Bioelementology, Yaroslavl State University, Sovetskaya St., 14, Yaroslavl 150000, Russia; 7Graduate Institute of Metabolism and Obesity Sciences, College of Nutrition, Taipei Medical University, Taipei 11031, Taiwan; 8Nutrition Research Center, Taipei Medical University Hospital, Taipei 11031, Taiwan; 9Chinese Taipei Society for the Study of Obesity (CTSSO), Taipei 11031, Taiwan

**Keywords:** soluble CD163, dietary pattern, AST, red blood cell aggregation, metabolic syndrome, obesity

## Abstract

Elevated soluble cluster of differentiation 163 (sCD163) concentrations, a marker of macrophage activation, are associated with obesity. Weight reduction decreases circulating CD163 levels, and changes in sCD163 levels are associated with improved metabolic dysfunction. Currently, the relationship between sCD163 and diet remains unclear. This study investigated dietary patterns associated with sCD163 concentrations and its predictive effect on metabolic syndrome (MetS). Data on anthropometrics, blood biochemistry, and a food frequency questionnaire were collected from 166 Taiwanese adults. sCD163 levels independently predicted MetS (odds ratio (OR): 5.35; 95% confidence interval (CI): 2.13~13.44, *p* < 0.001), non-alcoholic fatty liver disease (OR: 2.19; 95% CI: 1.03~4.64, *p* < 0.001), and central obesity (OR: 3.90; 95% CI: 1.78~8.55, *p* < 0.001), after adjusting for age and sex. An adjusted linear regression analysis revealed strong correlations between levels of sCD163 and aspartate transaminase (AST) (β = 0.250 (0.023~0.477), *p* < 0.05) and red blood cell aggregation (β = 0.332 (0.035~0.628), *p* < 0.05). sCD163-associated dietary pattern scores (high frequencies of consuming noodles and desserts, and eating at home, and a low intake frequency of steamed/boiled/raw food, white/light-green-colored vegetables, orange/red/purple-colored vegetables, dairy products, seafood, dark-green leafy vegetables, and soy products) were positively correlated with MetS, liver injury biomarkers, and sCD163 levels (all *p* for trend < 0.05). Individuals with the highest dietary pattern scores (tertile 3) had a 2.37-fold [OR: 2.37; 95% CI: 1.04~5.37, *p* < 0.05] higher risk of MetS compared to those with the lowest scores (tertile 1). Overall, the study findings suggest the importance of a healthy dietary pattern in preventing elevated sCD163 levels and diet-related chronic disease such as MetS.

## 1. Introduction

Substantial evidence has demonstrated a positive relationship between circulating soluble cluster of differentiation 163 (sCD163) concentrations and obesity-related comorbidities such as diabetes, non-alcoholic fatty liver disease (NAFLD), and metabolic syndrome (MetS) [1,2]. CD163 is a transmembrane scavenger receptor mainly expressed on monocyte-macrophage cell lineages. During macrophage activation, the CD163 surface receptor is shed as a soluble form by a metalloproteinase-dependent pathway, a mechanism that also involves cleavage of tumor necrosis factor (TNF)-α [3]. Other factors such as a high-fat diet, lipopolysaccharides, and endoplasmic reticular stress can also trigger CD163+ macrophage activation and the release of sCD163 [4,5,6]. Circulating sCD163 levels not only correlate with proinflammatory cytokines (e.g., TNF-α) but also the fat mass. It was found that circulating CD163 is positively correlated with messenger (m)RNA expression of CD163 in adipose tissues, and circulating CD163, and adipose CD163 mRNA levels are also correlated with the glucose disposal rate [7]. Hence, sCD163 levels might not only reflect the degree of macrophage infiltration into adipose tissues but also the ability of adipocytes to utilize glucose.

CD163+ macrophages also play an essential role in heme iron metabolism [4]. The CD163 receptor is responsible for recycling a cell′s free hemoglobin (Hb) via high-affinity binding with the Hb-haptoglobin complex. CD163+ macrophages can also take up senescent red blood cells (RBCs) and degrade heme iron. Within CD163+ macrophages, globin is degraded in lysosomes, and ferrous iron (Fe^2+^) is released from heme via heme oxygenase (HO)-1 [8]. CD163+ macrophage-RBC-Hb recycling pathways are important antioxidative pathways, as free Hb is a toxic pro-oxidant. Diet is known to affect the functions of RBCs and CD163+ macrophages. For example, animal studies showed that a diet high in fat increases RBC aggregation and deformability, and alterations of RBC membrane phospholipid components also activates macrophages [4,9].

Although diet and nutrients are known to modulate the activation status of CD163+ macrophages [4,9], questions remain as to the type of diet and nutrients that affect the shedding of the surface CD163 receptor from macrophages. Energy restriction-mediated weight reduction (e.g., bariatric surgery or a hypocaloric diet) decreased sCD163 levels, and changes in sCD163 levels were associated with improvements in metabolic dysfunction [1,10,11,12]. Currently, the direction of causal relationships among weight loss, sCD163, and metabolic improvement remains undefined [7]. sCD163 levels decreased in response to bariatric surgery [10], a hypocaloric diet [1,10], and lifestyle interventions (a combination of physical activities and caloric restrictions) [12] but not exercise [1] or physical activity [12]. Fjeldborg et al. showed that moderate intensity exercise without dietary restriction did not effect on sCD163 levels despite of losing 3.5% of their initial body weight [1]. A lifestyle intervention study involving 126 NAFLD patients also reported no correlation between sCD163 levels and total minutes of physical activity conducted by participants [12]. These studies imply that exercise alone may not have regulatory effects on the shedding of CD163 surface receptors from macrophages; in contrast, diet and nutrients can do so.

Currently, there is a limited number of publications investigating the relationship between dietary pattern and sCD163 levels. A better understanding of the relationship between diet and sCD163 levels is important for healthcare professionals to develop specific targeted dietary intervention programs for obese patients with elevated sCD163 levels. Understanding the role of diet in obesity-related comorbidities may also have clinical significance; intervention studies especially seem to suggest that diet, not exercise/physical activity [1,12], modulate sCD163 levels. To this end, we first investigated serum biomarkers that predict sCD163 levels and identified dietary pattern scores associated with circulating CD163 levels and their predictive effects on MetS in 166 Taiwanese adults.

## 2. Materials and Methods

### 2.1. Participants

A descriptive, cross-sectional study was conducted with a convenience sample of adults aged 20~64 years, who had previously attended the Taipei Medical University (TMU) Hospital from 1 June 2015 to 31 September 2015. This study was approved by the institutional ethical review committee of TMU (TMU-JIRB 201502018) and written informed consent was obtained from all participants. Exclusion criteria included the following: alcohol intake of >20 g/week for women or >30 g/week for men, disease history of cancer (e.g., hepatocarcinoma), nephritis, liver diseases (e.g., hepatitis virus infection, cholecystectomy), autoimmune disease, use of iron supplements in the past three months, and having missing data (e.g., blood samples, dietary intake data).

### 2.2. Definitions

The body weight, height, waist circumference (WC) and hip circumferences (HC) of each participant were recorded. The body mass index (BMI) was calculated as the weight divided by the height squared (kg/m^2^). The waist-hip (W/H) ratio was calculated by dividing the WC with the HC.

Type 2 diabetes was defined by subjects meeting at least one of the following criteria: (1) fasting plasma glucose (FPG) ≥126 mg/dL, (2) glycated hemoglobin (HbA1c) >6.5%, and (3) self-reported diabetes. Hyperlipidemia was diagnosed if subjects had full fill with at least one of the following criteria: (1) low-density lipoprotein cholesterol (LDL-C) ≥160 mg/dL, (2) high-density lipoprotein cholesterol (HDL-C) <40 mg/dL, (3) cholesterol (CHOL) ≥240 mg/dL, (4) triglyceride (TG) ≥200 mg/dL, and (5) a CHOL/HDL-C ratio ≥5 [13]. NALFD was diagnosed based on the results of ultrasound elastography scans and sonographic findings were evaluated by two independent experienced physicians [13]. MetS was defined based on the modified National Cholesterol Education Program Adult Treatment Panel III for the Asia Pacific: (1) WC in males ≥90 cm and females ≥80 cm, which is also the criteria for central obesity; (2) FPG ≥100 mg/dL; (3) HDL-C <40 mg/dL; (4) TG ≥150 mg/dL; and (5) systolic blood pressure ≥130 mmHg or diastolic blood pressure ≥85 mmHg. 

### 2.3. Questionnaires

Dietary data were collected by a validated self-administered food frequency questionnaire (FFQ) originally designed for the Nutrition and Health Survey in Taiwan [14]. Frequencies of eating out or eating at home and the type of cooking methods used to prepare food consumed by participants were also included to clarify the influence of life-style factors on obesity-related inflammation. The FFQ originally contained 64 food items, which were categorized into a total of 32 food groups. Response categories were as follows: (1) 0–1 time/week; (2) 2–3 times/week; (3) 4–5 times/week; (4) 6–7 times/week; (5) 8–10 times/week; (6) 11–13 times/week; (7) 14–16 times/week; and (8) ≥17 times/week. In addition, a self-reported medical health history questionnaire was also included. 

### 2.4. Anthropometric and Laboratory Measurements

Blood samples were collected from overnight fasted participants. Whole blood was used to analyze RBC rheology (deformability and aggregation). RBC deformability and aggregation was assessed using the RheoScan-AnD300 microfluidic ektacytometer (MicroStar Instruments, Seoul, Korea) [13]. A panel of biochemical biomarkers related to liver injury biomarkers (aspartate transaminase (AST), alanine transaminase (ALT), gamma-glutamyltransferase (γGT)), glucose (FPG, HbA1c, insulin, carboxymethyl-lysine (CML)) and lipid profiles (TG, HDL, LDL, CHOL), and iron biomarkers (serum ferritin (SF), iron, total iron binding capacity (TIBC)) were assessed. Serum transferrin saturation (% TS) was calculated as (serum iron ÷ TIBC) × 100%). Serum sCD163 (R & D Systems, Minneapolis, MN, USA) and free Hb (Immunology Consultants Laboratory, Portland, Oregon, USA) were measured by an enzyme-linked immunosorbent assay (ELISA) according to the manufacturer′s procedures. 

### 2.5. Statistical Analysis

Analyses were conducted using GraphPad Prism 5 (GraphPad Software, La Jolla, CA, USA), IBM SPSS 21 (IBM, Armonk, NY, USA), and SAS vers. 9.4 (SAS Institute, Cary, NC, USA). Categorical data are presented as the number and percentage (%), and continuous data are presented as the mean ± standard deviation (SD). A normality test was carried out to test for the distribution of each variable. Variables that were not normally distributed were log-transformed. A multivariate linear regression analysis was implemented to examine relationships of sCD163 with potential variables. A general linear model and Chi-squared test were respectively used to analyze the *p* for trend between variables for continuous data and categorical data. sCD163 data were divided into tertiles (Ts) with T1 being the smallest and T3 being the largest values (Table 1). Dietary pattern was constructed from the self-reported FFQ data using a data driven approach by a reduced rank regression (RRR) [15]. RRR was a new a posteriori method that extracted dietary pattern based on nutrients or biomarkers that have been associated with disease of interest [15]. RRR identifies linear functions of food groups that explain as much of the variation of selected nutrients or biomarkers; hence, RRR is considerably dependent on an adequate selection of response variables [16]. In this study, we selected response variables (RBC aggregation and serum AST) based on its strong relationship with sCD163 levels (Table 2). In addition, the 32 food groups from the FFQ were used as a predictor. The directed acyclic graph below explains the conceptual framework of the RRR (Figure 1). CD163-associated dietary pattern scores were selected with food groups that had factor loadings of ≥20 or ≤−0.20. Dietary pattern scores that were derived from each subject represented the sum of food intake variables weighted by the corresponding factor loading. These scores indicated the conformity of food consumption to the sCD163-associated dietary pattern. As suggested by Wsikert and Schulze [16], RRR is a modern statistical method used in nutritional epidemiology which can link dietary intake to biomarkers and test the hypothesis on pathways from diet to development of disease. Hence, the current study first investigated serum biomarkers that predict sCD163 levels, then linked sCD163-associated response variables to dietary intake to derive sCD163-related dietary pattern scores. Finally, the effects of sCD163-related dietary pattern scores on NAFLD and MetS were evaluated. A logistic regression model adjusted for age, sex, and log-transformed BMI was performed to estimate the odds ratio (OR) and 95% confidence interval (CI) of NAFLD and MetS. Differences were considered significant at *p* values of ≤0.05.

## 3. Results

The mean age of study subjects was 42.13 ± 12.61 years (men: 41.44 ± 11.79 years, women: 42.78 ± 13.35 years; *p* = 0.455), and BMI was 24.46 ± 5.06 kg/m^2^ (men: 25.50 ± 4.21 kg/m^2^, women: 23.50 ± 5.60 kg/m^2^; *p* < 0.001). Prevalence rates of central obesity, NAFLD, and MetS were 46.1% (men: 49.5%, women: 42.7%; *p* = 0.302), 81.3% (men: 79.3%, women: 83.2%; *p* = 0.447), and 24.4% (men: 26.2%, women: 22.8%; *p* = 0.561), respectively.

### 3.1. Serum sCD163 Levels Indepently Predict MetS

We next stratified individuals according to serum sCD163 concentrations into tertiles. Table 1 shows that sCD163 had positive trends with diabetes, dyslipidemia, NAFLD, MetS, BMI, liver injury markers (AST and ALT), TG, glucose biomarkers (HbA1c, insulin, and CML), RBC aggregation, and SF (all *p* for trend < 0.05). On the other hand, HDL-C had a negative trend with sCD163 concentrations (*p* for trend = 0.004).

Figure 2 shows that sCD163 levels independently predicted MetS (OR: 5.35; 95% CI: 2.13~13.44, *p* < 0.001), NAFLD (OR: 2.19; 95% CI: 1.03~4.64, *p* < 0.05), and central obesity (OR: 3.90; 95% CI: 1.78~8.55, *p* < 0.001), after adjusting for age and sex. The predictive effect of sCD163 levels on MetS remained significant after further adjusting for BMI (OR: 4.62; 95% CI: 1.58~13.51, *p* = 0.005) (data not shown).

### 3.2. Relationship between Serum sCD163 Levels and Potential Variables

We next performed a multivariate linear regression analysis to investigate potential confounding factors that were associated with serum sCD163 levels. Table 2 shows that after adjusting for covariates (log BMI, log AST, log TG, CML, log TIBC, and log Aggregation), only log-transformed AST (β = 0.250; 95% CI: 0.023~0.477, *p* = 0.031), and log-transformed RBC aggregation (β = 0.332; 95% CI: 0.035~0.628, *p* = 0.048) remained significantly correlated with serum sCD163 levels (model 2).

### 3.3. sCD163-Associated Dietary Pattern Scores by the RRR

sCD163-related dietary pattern scores were derived by the RRR from 32 food groups. Response variables were selected based on strong correlations between sCD163 and the independent variables, which were serum AST and RBC aggregation (both *p* < 0.05; model 2, Table 2). Table 3 shows that food groups of noodles and desserts, and the frequency of eating at home were positively correlated with the first dietary pattern scores (factor loadings of ≥0.20). On the other hand, steamed/boiled/raw food, white/light-green-colored vegetables, orange/red/purple-colored vegetables, dairy products, seafood, dark-green vegetables, and soy products were negatively correlated with dietary pattern scores (factor loadings of ≤−0.20).

### 3.4. Association between sCD163-Associated Dietary Pattern Scores and MetS

Dietary pattern scores were then stratified into tertile levels to investigate relationships between dietary pattern scores and potential variables. Table 4 shows that MetS (*p* < 0.05), liver injury biomarkers (AST and ALT) (both *p* < 0.001), insulin (*p* < 0.05), sCD163 levels (*p* < 0.001), and iron-related biomarkers (free Hb and hepcidin) (both *p* for trend < 0.05) had positive trends with dietary pattern scores. A small positive trend between central obesity index (WC and W/H ratio) and dietary pattern scores was also found (*p* for trend = 0.057 and 0.068, respectively). In contrast, dietary pattern scores were negatively correlated with HDL-C (*p* for trend = 0.009) (Table 4).

Table 5 shows that when compared to the lowest dietary pattern scores (T1 (Ref)), individuals with T3 scores (β = 0.235 (0.050~0.421), *p* = 0.013; model 3) were positively correlated with serum sCD163 levels after adjusting for age, sex, and log-transformed BMI (*p* for trend = 0.016; model 3).

We next performed a multivariate logistic regression to investigate the predictive effects of sCD163-associated dietary pattern scores on MetS and NAFLD. The multivariate logistic regression analysis showed that when compared to the lowest dietary pattern scores [T1 (Ref)], individuals with the highest dietary pattern scores (T3) had a 2.37-fold [OR: 2.37; 95% CI: 1.04~5.37, *p* < 0.05] higher risk of developing MetS after adjusting for age and sex (*p* for trend < 0.05) (Figure 3). However, there was no significant predictive effect of dietary pattern scores on NAFLD (*p* for trend = 0.17) (Figure 3).

## 4. Discussion

In agreement with previous studies [1,2], the current study found that sCD163 independently predicted central obesity, MetS, and NAFLD. In addition, the finding of the study further strengthened the importance of a healthy dietary pattern in preventing diet-related chronic disease such as elevated sCD163 levels and MetS. To put our results into perspective, we observed that one unit increased in dietary pattern scores, sCD163 levels increases 0.166 (β = 0.166 (0.051–0.281), *p* = 0.005) after adjusted for age, sex and BMI. However, as pointed out by Neuhouser ML [17], the exploratory data-driven approach like RRR-derived dietary pattern had high heterogeneity across study population. Hence, this limits us to provide a more clinical interpretation of the findings. Nonetheless, the current findings suggest that individuals adherence to the sCD163-related dietary pattern consumed lower frequency of vegetables and non-meat protein products (e.g., dairy products, seafood, and soy products) but higher intake of refined carbohydrates (noodles and desserts); hence, it is important to educate OW/obese individuals adopting a healthy diet. A high intake of vegetables and non-meat products and low intake of refined carbohydrates are cornerstones of a healthy diet, which is known to prevent cardiometabolic abnormalities [18,19].

The effects of diet/nutrients on sCD163 levels may involve both direct and indirect mechanisms. Overnutrition (e.g., fat, glucose, and iron) are known to trigger macrophage activation, which may, in turn, lead to direct shedding of the CD163 receptor from activated macrophages [4,6,9,20,21]. In contrast, nutrients like carotenoids [22] and isoflavones [23] are reported to attenuate macrophage polarization, and decrease cytokine release and systemic inflammation. The anti-inflammatory effects of carotenoids and isoflavones are supported by a recent human study in a Spanish population which showed that coffee and wine consumption frequencies were inversely correlated with circulating sCD163 levels [24].

To our surprise, the present study found that eating at home was associated with elevated serum sCD163 levels and risk of MetS. Our preliminary analysis found that compared to those with the lowest tertiles of frequency of eating at home, individuals with the highest frequency were associated with a younger age, higher sCD163 levels, and a higher prevalence rate of MetS (*p* for trend < 0.05; data not shown). In addition, participants in the highest tertile were also more likely to consume refined carbohydrates (e.g., bread and pastry, noodles, desserts, and fried desserts) western dishes, organ meats, processed meats, animal fats, beef and lamb, deep-fried foods, grilled/barbequed foods, and sea vegetables but less likely to consume fruits and white/light-green vegetables (*p* for trend < 0.05; data not shown). In contrast, older people had a lower frequency of eating at home and were more likely to have a better diet quality compared to younger people (eating at home tertile 1 vs. tertile 3:46.3 vs. 40.6 years old). This result is in line with a recent study investigating the association between the risk of obesity and at-home and away-from-home dietary patterns among Brazilian adolescents, which found that an unhealthy dietary pattern consumed at home was associated with an increased risk of obesity, while the away-from-home food consumption frequency was not [25]. Similar to their findings [25], the present study also found that participants who preferred to eat at home were more likely to have a Western dietary pattern and highlights the need to improve food choices when individuals prefer to eat at home.

sCD163-associated dietary pattern scores accounted for 75.41% of the total explained variation, with noodles and steamed/boiled/raw foods the strongest, at 13.87% and 16.01%, respectively. Although both RBC aggregation and AST were independent predictors of circulating sCD163, the contribution of RBC aggregation to dietary pattern scores was the lowest (explained variation: 4.83%) compared to AST (explained variation: 17.63%). Hence, a strong positive trend between sCD163-associated dietary pattern scores and liver injury (indicated by AST and ALT levels) was also observed (Table 4). In addition, individuals who were more likely to eat a high frequency of refined carbohydrates (noodles and desserts) but less steamed/boiled/raw foods, vegetables, and non-meat products were also more likely to develop elevated sCD163 levels and MetS. Noodles and desserts are refined carbohydrates with a high glycemic index and are strong risk factors for type 2 diabetes and MetS [19,26,27]. Moreover, a high intake of desserts also independently predicted MetS [27], and a high carbohydrate intake may also trigger dysregulated lipid and glucose metabolism [28]. A systematic review/meta-analysis suggested that a healthy dietary pattern, characterized by a high intake of vegetables, fresh fruits, whole grains, seafood, and low-fat dairy products, may protect against cancer and MetS possibly via its high content of vitamins, minerals, antioxidants, fiber, and n-3 fatty acids [18,29]. Overall, our data support the protective effect of vegetables and non-meat products against elevated risks of sCD163 and MetS. However, future studies need to investigate whether adopting a specific food pattern without caloric restriction can decrease sCD163 levels and improve metabolic function in subjects with elevated sCD163 levels.

In this study, AST and RBC aggregation were identified as independent predictors of sCD163 levels. Pearson’s correlation coefficients analysis showed that RBC aggregation was also positively correlated with AST (*r* = 0.202, *p* = 0.004) (data not shown). Relationships among dietary pattern scores, selected food groups, and selected responses (sCD163, AST, and RBC aggregation) were further analyzed to clarify which kind of food group was associated with which response. A univariate linear regression analysis suggested that log sCD163 (β = 0.211; 95% CI: 0.099~0.323, *p* < 0.001), log RBC aggregation (β = 0.057; 95% CI: 0.005~0.109, *p* = 0.033), and log AST (β = 0.138; 95% CI: 0.073~0.204, *p* < 0.001) were all positively correlated with dietary pattern scores (data not shown). After adjusting for age, sex, and log BMI, only log sCD163 (β = 0.166; 95% CI: 0.051~0.281, *p* = 0.005) and log AST (β = 0.097; 95% CI: 0.034~0.161, *p* = 0.034) remained significantly correlated with sCD163-associated dietary pattern scores (data not shown). Furthermore, a linear regression analysis showed that serum AST levels and RBC aggregation were strongly associated with eating at home (*p* = 0.003) and desserts (*p* = 0.06), respectively (data not shown). In contrast, sCD163 levels had a weak positive trend with noodles (p = 0.077) and negative trends with white/light-green-colored vegetables (*p* = 0.056) and steamed/boiled/raw food (*p* = 0.057) (data not shown). Human studies found that vegetables are inversely correlated with RBC aggregation [13], and increased consumption of green vegetable juice rich in dark-green leafy and cruciferous vegetables improves erythrocyte membrane function in patients with hypercholesterolemia [30]. Vegetables may improve RBC function by decreasing the n-6/n-3 ratio and saturated fatty acids and increasing polyunsaturated fatty acids in erythrocyte membrane phospholipids [30]. Overall, our results reemphasize the important roles of vegetables and plant products against macrophage-mediated chronic inflammation.

The major limitation of this study was the cross-sectional study design with a small sample size (*n* = 166), which was unable to distinguish cause and effect relationships. In this study, a dietary assessment was conducted by a self-reported FFQ and not standard dietary assessment methods (e.g., 24 h dietary recall) due to budget restraints and a lack of manpower. The FFQ is an instrument which is commonly used to investigate usual dietary intake and its relationship with health outcomes at population levels [31]. However, the current study did not assess the portion size of each food item; hence, the nutrient contents of individual food items could not be reported. Furthermore, the FFQ may be prone to bias as individuals with MetS may alter their habitual intake [32]. Although the sCD163-associated dietary pattern seemed to predict the risk of MetS, future studies need to investigate whether this dietary pattern is also relevant to other age groups (e.g., young adults and the elderly) as the mean age of our participants was middle-aged (42.13 ± 12.61 years). In addition, the sCD163-related dietary pattern was derived from an RRR based on interactions among selective response variables (sCD163, AST, and RBC aggregation) and food groups, and use of different response variables may yield different dietary patterns. Last but not the least, a randomized control trail is essential to test whether decreased consumption of refined carbohydrates (e.g., noodles and desserts) and increased intake of vegetables and non-meat products decrease circulating CD163 levels and improve metabolic dysfunction.

## 5. Conclusions

Relationships between sCD163 levels and obesity-related comorbidities (e.g., MetS) are complex, and dietary-related factors may serve as important mediators. The study results suggest that individuals who had lower intake frequencies of steamed/boiled/raw food, vegetables (dark leafy greens-, white/light greens-, orange/red/purple-colored vegetables), and non-meat products (dairy, seafood, and soy products) but preferred noodles, desserts, and eating at home were more likely to have elevated sCD163 levels, as well as the development of MetS. A follow-up or randomized controlled trial is needed to clarify the question of whether adopting specific food patterns affects sCD163 levels and if a change in sCD163 levels is associated with an improvement in metabolic function.

## Figures and Tables

**Figure 1 nutrients-11-00940-f001:**
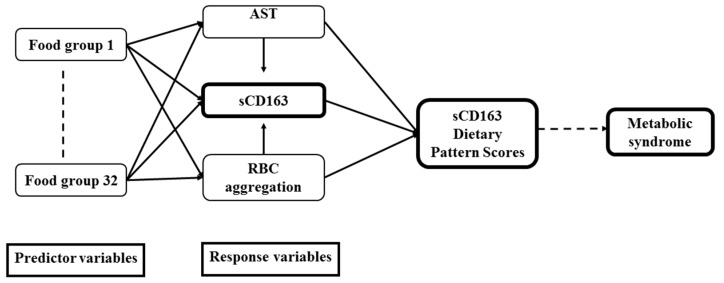
Directed acyclic graph of the reduced rank regression (RRR) conceptual framework. sCD163, soluble cluster of differentiation 163; AST, aspartate transaminase; RBCs, red blood cells.

**Figure 2 nutrients-11-00940-f002:**
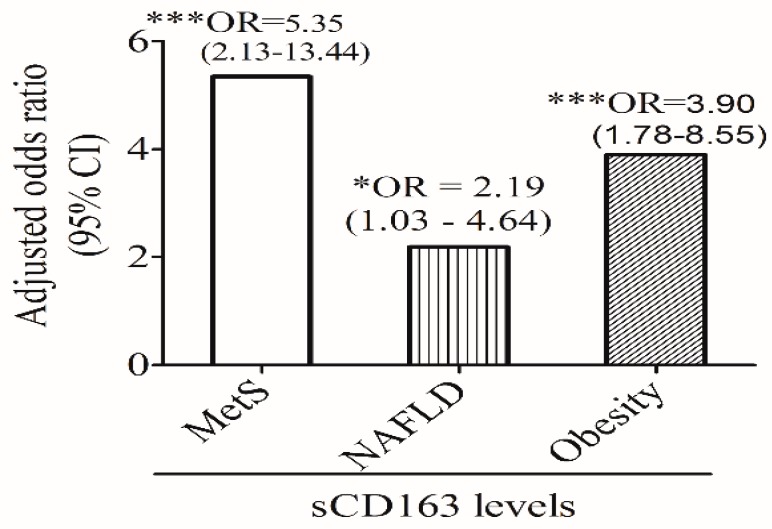
Age- and sex-adjusted odds ratios (ORs) and 95% confidence intervals (CIs) of soluble (s) CD163 levels for metabolic syndrome (MetS), non-alcoholic fatty liver disease (NAFLD), and central obesity. * *p* ≤ 0.05, *** *p* ≤ 0.001.

**Figure 3 nutrients-11-00940-f003:**
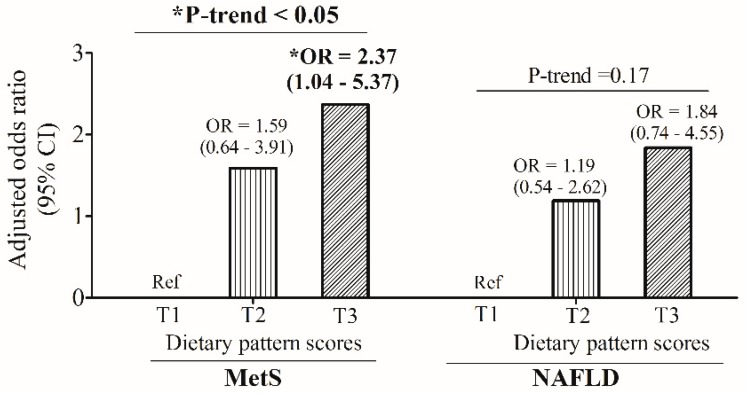
Age- and sex-adjusted odds ratios (ORs) and 95% confidence intervals (CIs) of dietary pattern score tertile levels for metabolic syndrome (MetS) and non-alcoholic fatty liver (NAFLD). * *p* ≤ 0.05.

**Table 1 nutrients-11-00940-t001:** Baseline characteristics of the study population according to tertiles of soluble (s)CD163 levels (*N* = 166).

Variable	sCD163, Tertiles ^$^			*p* for Trend *
	T_1_ (*n* = 55)	T_2_ (*n* = 56)	T_3_ (*n* = 55)	
**Basic Characteristics**				
Age (years)	40.40 ± 11.68	44.05 ± 13.20	42.19 ± 11.12	0.437
Male (*n*, %)	29 (52.7)	31 (55.4)	31 (56.4)	0.925
Diabetes (*n*, %)	4 (7.3)	4 (7.1)	13 (23.6)	**0.011**
Dyslipidemia (*n*, %)	14 (25.5)	24 (42.9)	28 (50.9)	**0.020**
Metabolic syndrome (*n*, %)	6 (10.9)	17 (30.4)	22 (40.0)	**0.002**
NAFLD (*n*, %)	36 (65.5)	46 (82.1)	49 (89.1)	**0.008**
**Anthropometrics**				
Waist circumference (cm)	81.17 ± 9.93	90.12 ± 13.87	90.26 ± 15.08	**<0.001**
Waist/hip ratio	0.83 ± 0.06	0.88 ± 0.08	0.88 ± 0.09	**0.001**
Body-mass index (kg/m^2^)	22.73 ± 3.84	25.99 ± 5.76	25.88 ± 5.88	**0.002**
**Liver injury biomarkers**				
Aspartate transaminase (U/L)	21.58 ± 6.73	25.98 ± 9.18	33.76 ± 24.75	**<0.001**
Alanine transaminase (U/L)	24.91 ± 16.79	32.88 ± 20.17	48.11 ± 39.31	**<0.001**
γ-Glutamyltransferase (U/L)	22.65 ± 16.74	25.68 ± 16.77	42.27 ± 87.62	0.051
**Lipid biomarkers**				
Total cholesterol (mg/dL)	202.33 ± 37.70	195.91 ± 35.25	198.22 ± 39.73	0.567
Triglycerides (mg/dL)	91.05 ± 45.06	130.54 ± 79.83	151.75 ± 91.92	**<0.001**
HDL-C (mg/dL)	61.79 ± 17.14	52.79 ± 13.01	52.86 ± 17.10	**0.004**
LDL-C (mg/dL)	119.02 ± 31.81	118.52 ± 33.43	117.07 ± 32.08	0.754
**Glucose-related biomarkers**				
FPG (mg/dL)	89.04 ± 13.41	91.27 ± 16.57	95.11 ± 24.81	0.093
HbA1c (%)	5.64 ± 0.50	5.83 ± 1.08	6.14 ± 1.36	**0.012**
Insulin (μIU/mL)	8.71 ± 6.00	11.16 ± 6.54	11.69 ± 7.50	**0.021**
CML (μg/mL)	225.46 ± 133.36	233.88 ± 149.22	301.59 ± 99.15	**0.003**
**Red blood cells (RBCs) and iron biomarkers**				
sCD163 (ng/mL)	457.24 ± 114.64	753.71 ± 76.02	1300.93 ± 506.57	**<0.001**
HCT (%)	44.35 ± 9.24	43.06 ± 5.89	43.05 ± 6.88	0.361
RBCs (MIL/mm^3^)	5.20 ± 1.15	5.07 ± 0.75	5.01 ± 0.94	0.314
Hemoglobin (Hb) (g/dL)	15.19 ± 3.26	14.82 ± 2.25	14.82 ± 2.52	0.465
Free Hb (μg/mL)	149.16 ± 52.22	162.91 ± 53.35	155.12 ± 50.44	0.562
Fe (μg/dL)	110.80 ± 38.16	99.57 ± 33.21	112.25 ± 41.42	0.840
TIBC (μg/dL)	348.70 ± 46.58	359.93 ± 49.56	364.15 ± 50.17	0.099
SF (ng/mL)	130.47 ± 125.02	160.47 ± 131.27	208.72 ± 192.96	**0.008**
TS (%)	32.45 ± 11.78	28.35 ± 10.66	31.59 ± 13.19	0.704
Hepcidin (ng/mL)	137.31 ± 106.09	157.45 ± 126.14	176.57 ± 103.13	0.071
Deformability SS^1/2^ (Pa)	2.21 ± 0.22	2.23 ± 0.21	2.25 ± 0.18	0.273
RBC aggregation CSS (mPa)	263.07 ± 67.24	302.82 ± 91.64	319.17 ± 109.69	**0.002**

* *p* for trend was analyzed by a general linear model for continuous variables and Chi-squared test for categorical variables. Continuous data are presented as the mean ± standard deviation, while categorical data are presented as the number (percentage of the same group). ^$^ sCD163 tertiles: tertile 1 ≤ 623.43 ng/mL, tertile 2 > 623.43~ ≤ 890.99, tertile 3 > 890.99. TS, transferrin saturation; NAFLD, non-alcoholic fatty liver disease; HDL-C, high-density lipoprotein cholesterol; LDL-C, low-density lipoprotein cholesterol; HCT, hematocrit, Fe, serum iron; SF, serum ferritin; sCD163, soluble cluster of differentiation 163; SS, shear stress; Pa, Pascal; CSS, critical shear stress; mPa, milliPascal; TIBC, total iron-binding capacity; CML, N-(carboxymethyl)lysine.

**Table 2 nutrients-11-00940-t002:** Multivariate linear regression of soluble CD163 levels and selected anthropometric, inflammation, lipid, glucose, and iron-related biomarkers (*N* = 166).

		Univariate		Model 1 *		Model 2 ^#^	
		β (95% CI)	*p* Value	β (95% CI)	*p* Value	β (95% CI)	*p* Value
Age (years)		0.005(−0.002~0.012)	0.152				
Sex							
	Male	Ref					
	Female	0.042 (−0.123~0.207)	0.616				
**Anthropometrics**							
	Log waist circumference (cm)	0.009(0.003~0.015)	**0.002**				
	Waist/hip ratio	1.442(0.452~2.433)	**0.005**				
	Log Body-mass index (kg/m^2^)	0.636 (0.233~1.039)	**0.002**	0.636 (0.233~1.039)	**0.002**	0.166 (−0.278~0.609)	0.462
**Liver injury biomarkers**							
	Log Aspartate transaminase (U/L)	0.391 (0.188~0.593)	**<0.001**	0.313 (0.096~0.530)	**0.005**	0.250 (0.023~0.477)	**0.031**
	Log Alanine transaminase (U/L)	0.198 (0.073~0.322)	**0.002**	0.135 (−0.005~0.275)	0.059		
	Log γ-Glutamyltransferase (U/L)	0.185 (0.062~0.307)	**0.003**	0.142 (0.017~0.268)	**0.027**		
**Lipid biomarkers**							
	Total cholesterol (mg/dL)	−0.002(−0.004~ < 0.001)	0.080				
	Log Triglycerides (mg/dL)	0.261 (0.135~0.388)	**<0.001**	0.213 (0.073~0.353)	**0.003**	0.086 (−0.063~0.236)	0.254
	Log HDL-C (mg/dL)	−0.483 (−0.779~−0.187)	**0.002**	−0.335 (−0.671~0.001)	0.051		
	LDL-C (mg/dL)	−0.002(−0.004~0.001)	0.181				
**Glucose-related biomarkers**							
	Log FPG (mg/dL)	0.356 (−0.134~0.846)	0.153				
	Log HbA1c (%)	0.578 (0.022~1.134)	**0.042**	0.394 (−0.168~0.956)	0.168		
	Log Insulin (µIU/mL)	0.180 (0.032~0.327)	**0.018**	0.070 (−0.11~0.249)	0.444		
	CML (μg/mL)	0.001 (0.00026~0.002)	**0.006**	0.001 (0.000129~0.00138)	**0.019**	0.001 (−0.00005~0.001)	0.072
**RBCs and iron biomarkers**							
	Log HCT (%)	−0.113 (−0.618~0.392)	0.658				
	Log RBC (MIL/mm^3^)	−0.147 (−0.602~0.308)	0.525				
	Log Hb (g/dL)	−0.106 (−0.582~0.370)	0.661				
	Free Hb (μg/mL)	0.00038 (−0.001~0.002)	0.655				
	Fe (μg/dL)	0.00028 (−0.002~0.002)	0.799				
	Log TIBC (µg/dL)	0.661 (0.068~1.253)	**0.029**	0.583 (0.002~1.164)	**0.049**	0.329 (−0.264~0.923)	0.274
	Log SF (ng/mL)	0.053 (−0.018~0.125)	0.142				
	TS (%)	−0.002 (−0.009~0.005)	0.604				
	Hepcidin (ng/mL)	0.001 (−0.000115~0.001)	0.097				
	Deformability SS1/2 (Pa)	0.136 (−0.268~0.541)	0.507				
	Log RBC aggregation CSS (mPa)	0.541 (0.259~0.824)	**<0.001**	0.473 (0.190~0.756)	**0.001**	0.332 (0.035~0.628)	**0.029**

* Model 1: Adjusted for log BMI; ^#^ Model 2: adjusted for log BMI, log AST, log TG, CML, Log TIBC, log Aggregation CSS. Abbreviations are defined in the footnotes to Table 1.

**Table 3 nutrients-11-00940-t003:** Food groups that were strongly associated with the soluble CD163-related dietary pattern scores identified by the reduced rank regression (RRR).

Figure	Explained Variation (%)	Factor Loading *
**Positive association**		
Noodles	13.87	0.38
Eating at home	6.09	0.26
Desserts	5.87	0.25
**Negative association**		
Steamed/boiled/raw food	16.01	−0.41
White/light-green-colored vegetables	8.14	−0.29
Orange/red/purple-colored vegetables	6.46	−0.26
Dairy products	5.19	−0.24
Seafood	5.25	−0.24
Dark-green vegetables	4.42	−0.22
Soy products	4.10	−0.21
**Total:**	75.41	

* Factor loadings are correlations between food groups and the first dietary pattern scores (correlation coefficient for the RRR-derived pattern ≥|0.20|).

**Table 4 nutrients-11-00940-t004:** Characteristics of the study population according to tertiles of dietary pattern scores.

Variable		sCD163-Related Dietary Pattern Scores, Tertile ^$^			*p* for Trend *
		T_1_ (*n* = 57)	T_2_ (*n* = 56)	T_3_ (*n* = 52)	
**Basic characteristic**					
	Age (years)	41.87 ± 13.29	39.70 ± 12.66	45.13 ± 11.34	0.108
	Male (*n*, %)	36 (47.4)	42 (55.3)	33 (43.3)	0.331
	Diabetes (*n*, %)	8 (11.3)	8 (10.5)	10 (13.7)	0.823
	Dyslipidemia (*n*, %)	23 (32.4)	27 (35.5)	33 (45.2)	0.252
	Metabolic syndrome (*n*, %)	12 (16.9)	16 (21.1)	25 (34.2)	**0.039**
	NAFLD (*n*, %)	59 (77.6)	60 (78.9)	67 (88.2)	0.189
**Anthropometrics**					
	Waist circumference (cm)	83.43 ± 10.49	86.54 ± 14.34	87.49 ± 13.88	**0.057**
	Waist/hip ratio	0.85 ± 0.07	0.86 ± 0.08	0.87 ± 0.09	0.068
	Body-mass index (kg/m^2^)	23.65 ± 3.58	24.83 ± 5.93	24.81 ± 5.21	0.154
**Inflammation biomarkers**					
	Aspartate transaminase (U/L)	22.27 ± 6.39	24.84 ± 10.27	31.62 ± 22.05	**<0.001**
	Alanine transaminase (U/L)	24.79 ± 13.05	31.49 ± 26.31	40.56 ± 33.38	**<0.001**
	γ-Glutamyltransferase (U/L)	23.77 ± 18.35	23.79 ± 20.25	34.74 ± 75.97	0.159
**Lipid biomarkers**					
	Total cholesterol (mg/dL)	206.13 ± 35.14	196.05 ± 38.04	195.34 ± 38.21	0.083
	Triglycerides (mg/dL)	106.51 ± 69.25	117.76 ± 80.16	128.96 ± 72.23	0.071
	HDL-C (mg/dL)	61.17 ± 16.47	55.81 ± 15.61	54.32 ± 14.42	**0.009**
	LDL-C (mg/dL)	122.44 ± 32.80	115.97 ± 32.29	115.74 ± 30.82	0.210
**Glucose-related biomarkers**					
	FPG (mg/dL)	92.61 ± 19.99	88.43 ± 13.80	93.36 ± 20.62	0.806
	HbA1c (%)	5.89 ± 1.07	5.64 ± 0.57	5.95 ± 1.13	0.716
	Insulin (μIU/mL)	8.64 ± 4.23	10.45 ± 7.63	10.93 ± 6.05	**0.027**
	CML (μg/mL)	242.81 ± 141.57	271.81 ± 142.55	258.05 ± 114.08	0.550
**Red blood cells (RBCs) and iron biomarkers**					
	sCD163 (ng/mL)	685.04 ± 299.98	851.39 ± 403.14	996.00 ± 597.75	**<0.001**
	HCT (%)	43.99 ± 8.14	43.16 ± 6.37	41.85 ± 6.87	0.074
	RBCs (MIL/mm^3^)	5.11 ± 0.99	5.06 ± 0.84	4.95 ± 0.88	0.305
	Hb (g/dL)	14.97 ± 2.89	14.85 ± 2.44	14.35 ± 2.57	0.160
	Free Hb (μg/mL)	140.68 ± 50.46	154.15 ± 52.22	165.62 ± 54.14	**0.013**
	Fe (μg/dL)	104.37 ± 38.07	101.97 ± 37.82	106.32 ± 37.52	0.757
	TIBC (μg/dL)	360.25 ± 53.58	356.71 ± 56.31	367.38 ± 52.70	0.432
	SF (ng/mL)	119.95 ± 111.11	158.54 ± 145.60	164.15 ± 175.26	0.072
	TS (%)	29.55 ± 11.31	29.67 ± 12.60	29.93 ± 12.32	0.850
	Hepcidin (ng/mL)	132.29 ± 100.45	146.14 ± 102.16	171.58 ± 114.43	**0.029**
	Deformability SS^1/2^ (Pa)	2.22 ± 0.20	2.21 ± 0.20	2.23 ± 0.21	0.745
	RBC aggregation CSS (mPa)	283.76 ± 65.16	294.69 ± 83.17	304.94 ± 118.31	0.187

* *p* for trend was analyzed by a general linear model for continuous variables and Chi-squared test for categorical variables. Continuous data are presented as the mean ± standard deviation, while categorical data are presented as the number (percentage of the same group). ^$^ Dietary pattern score tertiles: tertile 1 ≤−0.2527; tertile 2 > −0.2527~ ≤ 0.3240; tertile 3 > 0.3240. Abbreviations are defined in the footnotes to Table 1.

**Table 5 nutrients-11-00940-t005:** Linear regression of the relationship between tertiles of dietary pattern scores and log-transformed sCD163 levels.

	Dietary Pattern Scores ^$^					*p* for Trend
	T 1	T2	*p* Value	T 3	*p* Value	
**Univariate**	Ref	0.185 (−0.007~0.376)	0.059	0.320 (0.138~0.501)	**0.001**	**0.002**
**Model 1 ***	Ref	0.189 (−0.003~0.381)	0.054	0.293 (0.110~0.475)	**0.002**	**0.003**
**Model 2 ^#^**	Ref	0.201 (0.006~0.395)	**0.043**	0.298 (0.116~0.480)	**0.002**	**0.003**
**Model 3 ^$^**	Ref	0.165 (−0.028~0.358)	0.092	0.235 (0.050~0.421)	**0.013**	**0.016**

* Model 1: adjusted for age; ^#^ Model 2: adjusted for age and sex; ^$^ Model 3: adjusted for age, sex, and log body-mass index. ^$^ Dietary pattern score tertiles: tertile 1 ≤−0.2527; tertile 2 >−0.2527~ ≤0.3240; tertile 3 >0.3240.

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
