# Peer review of "Soluble CD163-Associated Dietary Patterns and the Risk of Metabolic Syndrome"

_nutrients, 2019, doi:10.3390/nu11040940_

Round 1
Reviewer 1 Report
Overall, this correlation study investigated the associated relationship between dietary habits and circulating CD163 and discussed the possibility to use sCD163 as a predictor for the risk of metabolic syndrome. The paper was well written and the experiments are well-designed. Thus, the manuscript should be accepted for publication after address the following comments.
1. Please define the sCD163-related dietary pattern score. Is there a method to calculate the food intake of carbohydrate, fat, proteins etc instead of the food types? It should be make more sense to study the effect of high sugar or high fat diet on secreted sCD163 levels.
2. Dose sCD163 level has age or sex difference? Did the author analyze the data by separating male and female?
3. Please explain why the sample size for Table1 (n=56) and Table 4 (n=76) are different.
Author Response
Reviewer 1 comments
1. Please define the sCD163-related dietary pattern score. Is there a method to calculate the food intake of carbohydrate, fat, proteins etc instead of the food types? It should be make more sense to study the effect of high sugar or high fat diet on secreted sCD163 levels.
We total agree with your commend in which that dietary fat and sugar may also affect sCD163 levels. Unfortunately, this study attempt to focus on the relationship between dietary pattern that associates with CD163 levels; hence, we only used FFQ to collect dietary information and didn’t assess nutrient intake. However, we can evaluate the effects of nutrient intake on sCD163 levels in our future study.
2. Dose sCD163 level has age or sex difference? Did the author analyze the data by separating male and female?
Thank you for your commend. As in the following table, we observed no age or sex difference in our study population. That’s why we did not separate male and female in the following analysis.
Variable | Sex | P-value | ||
Male | Female | |||
Basic characteristic | ||||
Age (yrs) | 41.44 ± 11.79 | 42.78 ± 13.35 | 0.455 | |
Diabetes (n, %) | 14(13.1) | 12(10.5) | 0.555 | |
Dyslipidemia (n, %) | 41(38.3) | 43(37.7) | 0.927 | |
MetS (n, %) | 28(26.2) | 26(22.8) | 0.561 | |
sCD163 (ng/mL) | 839.95 ± 487.96 | 832.96 ± 428.88 | 0.838 | |
3. Please explain why the sample size for Table1 (n=56) and Table 4 (n=76) are different
Sorry for the mistake. The total number was only 166 samples so we have changed number in Table 4 to n=56. (page 10, table 4)

Reviewer 2 Report
I would like to see a more detailed introduction.
I would have appreciated if the authors talked about the significance of the research and this could be manifested into translational medicine.
--------------------------------------------------------------------------------
Pros: The authors have explained the methods of the paper in a very detailed manner and have shown that" there is a bidirectional relationship between sCD163 and diet" .
Cons: In the introduction, the Authors have stated all the facts, however, they have failed to paint the bigger picture as to what is the relevance of their hypothesis and their conclusions. They have failed to tell a story and have failed to draw a conclusion stating the clinical significance of their finding, which questions the very novelty and significance of their work.
"Our results suggest a bidirectional relationship between diet and sCD163 levels, and changes in dietary patterns may modulate the relationship between sCD163 levels and the risk of MetS."= Their conclusion
Only if the authors address the above concerns, it may be published.
I also found a few minor errors
Introduction:
1) Fjeldborg et al. showed that moderate intensive exercise without dietary restriction had no effect -Intensity
2) exercise without dietary restriction had no effect on sCD163 - did not effect
3) in spite of participants losing 3.5% of their initial body weight [1]- despite
4) correlated with messenger (m)RNA expression of CD163 in adipose tissues, and circulating CD163, and adipose CD163 mRNA levels- add a comma
5) CD163+ macrophages also play an important role in heme iron metabolism [8].- essential
6) It is well accepted that obesity-related inflammation contributes to MetS and diet/nutrientinflammation interactions may play a key role in this relationship [13-15]. -nutrient inflammation add space and replace key with vital
Discussion:
7)Noodles and desserts are refined carbohydrates with a high glycemic index, and are strong risk factors for type 2 diabetes- remove comma
8) Vegetables may improve RBC function through decreasing the n-6/n-3 ratio and - replace with by
9) Although the sCD163-associated dietary pattern seemed to predict risk of MetS, future studies need to investigate- the risk
10) The study results suggest that individuals who had lower intake frequencies of steamed/boiled/raw food, vegetables (dark leafy green-, white/light green-, orange/red/purple-colored vegetables), and non-meat products (dairy, seafood, and soy products) but preferred noodles, disserts, and eating at home were more likely to
have elevated sCD163 levels, as well as the development of MetS. - replace with greens and correct the spelling of desserts
Author Response
Reviewer 2 comments
I would like to see a more detailed introduction. I would have appreciated if the authors talked about the significance of the research and this could be manifested into translational medicine.
Pros: The authors have explained the methods of the paper in a very detailed manner and have shown that" there is a bidirectional relationship between sCD163 and diet" .
Cons: In the introduction, the Authors have stated all the facts, however, they have failed to paint the bigger picture as to what is the relevance of their hypothesis and their conclusions. They have failed to tell a story and have failed to draw a conclusion stating the clinical significance of their finding, which questions the very novelty and significance of their work.
"Our results suggest a bidirectional relationship between diet and sCD163 levels, and changes in dietary patterns may modulate the relationship between sCD163 levels and the risk of MetS."= Their conclusion. Only if the authors address the above concerns, it may be published.
Thank you for pointing out the inclarity in the introduction, conclusion and discussion.
(1) We have revised the introduction to make the aim of the study more clear.
“Although diet and nutrients are known to modulate the activation status of CD163+ macrophages [8,12], questions remain as to the type of diet and nutrients that affect the shedding of the surface CD163 receptor from macrophages. Energy restriction-mediated weight reduction (e.g., bariatric surgery or a hypocaloric diet) decreased sCD163 levels, and changes in sCD163 levels were associated with improvements in metabolic dysfunction [1,3-5]. Currently, the direction of causal relationships among weight loss, sCD163, and metabolic improvement remains undefined [6]. sCD163 levels decreased in response to bariatric surgery [3], a hypocaloric diet [1,3], and lifestyle interventions (a combination of physical activities and caloric restrictions) [5] but not exercise [1] or physical activity [5]. Fjeldborg et al. showed that moderate intensity exercise without dietary restriction did not effect on sCD163 levels despite of losing 3.5% of their initial body weight [1]. A lifestyle intervention study involving 126 NAFLD patients also reported no correlation between sCD163 levels and total minutes of physical activity conducted by participants [5]. These studies imply that exercise alone may not have regulatory effects on the shedding of CD163 surface receptors from macrophages; in contrast, diet and nutrients can do so.
Currently, there is limited publication investigating the relationship between dietary pattern and sCD163 levels. A better understanding of the relationship between diet and sCD163 levels is important for healthcare professionals to develop specific targeted dietary intervention programs for obese patients with elevated sCD163 levels. Understanding the role of diet in obesity-related comorbidities may also have clinical significance especially interventions studies seem to suggest that diet, not exercise/physical activity [1,5], modulate sCD163 levels. To this end, we first investigated serum biomarkers that predict sCD163 levels and identified dietary pattern scores associated with circulating CD163 levels and their predictive effects on MetS in 166 Taiwanese adults.” (page 2-3, introduction, third and fourth paragraphs).
(2) We have changed conclusion in the abstract to “Overall, the study findings suggest the importance of healthy dietary pattern in preventing elevated sCD163 levels and diet-related chronic disease such as MetS.” (page 1).
(3) We have revised Discussion to provide a more clinical interpretation of the finding.
“In agreement with previous studies [1,2], the current study found that sCD163 independently predicted central obesity, MetS, and NAFLD. In addition, the finding of the study further strength the importance of healthy dietary pattern in preventing diet-related chronic disease such as elevated sCD163 levels and MetS. To point our results into perspective, we observed that one unit increased in dietary pattern scores, sCD163 levels increases 0.166 [ß=0.166 (0.051-0.281), p=0.005] after adjusted for age, sex and BMI. However, as pointed out by Neuhouser ML [21], the exploratory data-driven approach like RRR-derived dietary pattern had high heterogeneity across study population. Hence, this limits us to provide a more clinical interpretation of the findings. Nonetheless, the current findings suggest that individuals adherence to the sCD163-related dietary pattern consumed lower frequency of vegetables and non-meat protein products (e.g., dairy products, seafood, and soy products) but higher intake of refined carbohydrates (noodles and desserts); hence, it is important to educate OW/obese individuals adopting a healthy diet. A high intake of vegetables and non-meat products and low intake of refined carbohydrate are cornerstones of a healthy diet, which is known to prevent cardiometabolic abnormalities [22,23].” (page 12-13, Discussion: first paragraphs).
I also found a few minor errors
Introduction:
1) Fjeldborg et al. showed that moderate intensive exercise without dietary restriction had no effect –Intensity
Thank you so much. We have corrected it. (page 3, Introduction).
2) exercise without dietary restriction had no effect on sCD163 - did not effect
Thank you so much. We have corrected it. (page 3, Introduction).
3) in spite of participants losing 3.5% of their initial body weight [1]- despite
Thank you so much. We have corrected it. (page 3, Introduction).
4) correlated with messenger (m)RNA expression of CD163 in adipose tissues, and circulating CD163, and adipose CD163 mRNA levels- add a comma
Thank you so much. We have corrected it. (page 2, Introduction).
5) CD163+ macrophages also play an important role in heme iron metabolism [8].- essential
Thank you so much. We have corrected it. (page 2, Introduction).
6) It is well accepted that obesity-related inflammation contributes to MetS and diet/nutrientinflammation interactions may play a key role in this relationship [13-15]. -nutrient inflammation add space and replace key with vital
Thank you so much. We have removed this paragraph after revising the introduction.
Discussion:
7)Noodles and desserts are refined carbohydrates with a high glycemic index, and are strong risk factors for type 2 diabetes- remove comma
Thank you so much. We have corrected it. (page 13, Discussion).
8) Vegetables may improve RBC function through decreasing the n-6/n-3 ratio and - replace with by
Thank you so much. We have corrected it. (page 14, Discussion).
9) Although the sCD163-associated dietary pattern seemed to predict risk of MetS, future studies need to investigate- the risk
Thank you so much. We have corrected it. (page 14, Discussion).
10) The study results suggest that individuals who had lower intake frequencies of steamed/boiled/raw food, vegetables (dark leafy green-, white/light green-, orange/red/purple-colored vegetables), and non-meat products (dairy, seafood, and soy products) but preferred noodles, disserts, and eating at home were more likely to have elevated sCD163 levels, as well as the development of MetS. - replace with greens and correct the spelling of desserts
Thank you so much. We have corrected it. (page 15, Discussion).

Reviewer 3 Report
The manuscript “Soluble CD163-Associated Dietary Patterns and the Risk of Metabolic Syndrome” is a cross-sectional study that relates levels of CD163 and a dietary questionaire with biochemical and anthropometric variables of the participants. The topic is interesting as CD163 could be used as a biomarker of metabolic syndrome, however, the design of the study has some limitations as well as the statistic approaches used throughout the document are of difficult follow-up. This reviewer has several concerns.
The RRR must be better explained as it is difficult to know how it was addressed and it is of capital importance for the results.
The authors should share the FFQ used in order to better comprehend the data used in the study.
One of the main results is the fact that eating at home was associated with elevated serum CD163 levels and risk of MetS. As the authors stated, this result is very surprising, as it is expected the contrary. It seems like the age of the subjects is a main reason. Thus, in my opinion, this variable must be re-codified and calculated taking into account more information to better comprehend the results, as it seems like the way it has been calculated is not the proper way.
Another important result centers on AST and ALT levels, are they on pathogenic levels according to your population?
The different diseases included throughout the manuscript: metabolic syndrome, NAFLD and obesity confuse to the reader.
A dietary intervention or weight loss trial could indicate the real importance of the results.
Author Response
Thank you so much. We have revised by making a track change version of the manuscript.
